# Therapeutic and Prophylactic Use of Oral, Low-Dose IFNs in Species of Veterinary Interest: Back to the Future

**DOI:** 10.3390/vetsci8060109

**Published:** 2021-06-11

**Authors:** Sara Frazzini, Federica Riva, Massimo Amadori

**Affiliations:** 1Gastroenterology and Endoscopy Unit, Fondazione IRCCS Cà Granda, Ospedale Maggiore Policlinico, 20122 Milan, Italy; sara.frazzini@policlinico.mi.it; 2Dipartimento di Medicina Veterinaria, Università degli Studi di Milano, 26900 Lodi, Italy; 3Rete Nazionale di Immunologia Veterinaria, 25125 Brescia, Italy; m_amadori@fastwebnet.it

**Keywords:** veterinary immunotherapy, cytokines, IFN, low dose treatment, oral treatment

## Abstract

Cytokines are important molecules that orchestrate the immune response. Given their role, cytokines have been explored as drugs in immunotherapy in the fight against different pathological conditions such as bacterial and viral infections, autoimmune diseases, transplantation and cancer. One of the problems related to their administration consists in the definition of the correct dose to avoid severe side effects. In the 70s and 80s different studies demonstrated the efficacy of cytokines in veterinary medicine, but soon the investigations were abandoned in favor of more profitable drugs such as antibiotics. Recently, the World Health Organization has deeply discouraged the use of antibiotics in order to reduce the spread of multi-drug resistant microorganisms. In this respect, the use of cytokines to prevent or ameliorate infectious diseases has been highlighted, and several studies show the potential of their use in therapy and prophylaxis also in the veterinary field. In this review we aim to review the principles of cytokine treatments, mainly IFNs, and to update the experiences encountered in animals.

## 1. Introduction

Cytokines are biological molecules belonging to a group of low-molecular weight proteins or glycoproteins, synthesized and secreted, mainly by T helper (Th) lymphocytes and macrophages, in response to inducing stimuli [1]. They play an important role in various biological processes [2] and are able to regulate cell functions in a coordinated and interactive way thanks to different characteristics, such as pleiotropy, synergy, redundancy, antagonism and cascade induction [3]. Thanks to their properties, cytokines play an important role in maintaining and regulating the immune system. Sometimes the balance between humoral- and cell-mediated immune responses can break down due to an imbalance in cytokine production and/or cytokine receptor activation, leading to the onset of various pathological disorders [4]. In the last few years, researchers have investigated the therapeutic role of cytokines, also highlighting the anti-inflammatory properties of certain cytokines such as the interferons (IFNs) [5]; this led to the development of new treatments for both infectious and non-infectious disorders, such as AIDS, Hepatitis B, mycobacterial infections, anemia, rheumatoid arthritis, inflammatory dermatological diseases, interstitial pneumonia, allergic encephalomyelitis and neutropenia [6]. Moreover, cytokines can be used as immunomodulators in order to increase the host’s defenses by providing supra-physiological dosages of such mediators [7]. Interestingly, it has been shown that during inflammation there is a link between the IFN response and the tristetraprolin-mediated decay of mRNAs, thus highlighting the role IFNs play in the homeostasis of the immune system [8]. Recent studies, carried out on animal tumor models, show that different cytokines, including GM-CSF, IL-7, IL-12, IL-15, IL-18 and IL-21, are also able to limit tumor growth [9]. This important role in the regulation of the immune response outlines a potential use of cytokines as immuno-adjuvants for vaccination [10]. In fact, the addition of cytokines, as recombinant proteins or as cytokine-coded plasmids, causes immunomodulation or, more simply, an enhanced immune response to vaccines [6]. For example, the addition of recombinant IFNs to a flu vaccine has been shown to increase the efficacy of that vaccine [11]. Cytokines have also been investigated in veterinary medicine, at first for diagnostic purposes, and then as a valid alternative to the large use of antibiotics in farm animals, in order to reduce resistance to antibiotics also used in human medicine [12]. In particular, in vivo experiments have demonstrated the therapeutic efficacy of using IFN-α to treat viral infections, such as respiratory tract infections in cattle, transmissible gastroenteritis or rotavirus diarrhea in pigs, and other viral infections in mice, as well as immune-mediated diseases, such as polymyositis, multiple sclerosis and experimental autoimmune allergic encephalomyelitis in rats and mice [13,14]. The use of low-dose oral IFN-α also proved useful in the treatment of cats with feline leukemia virus (FeLV) and feline immunodeficiency virus (FIV), leading to a dramatic improvement in the health conditions of the tested subjects [15,16].

In addition, the availability of cytokine reagents has also enabled the development of new effective non-rodent animal models, such as sheep, for the study of human diseases [17]. To this end, the use of low-dose IFN treatments in livestock, pets and laboratory animals will be analyzed in detail.

## 2. Parenteral and Oromucosal Administration of Cytokines

The administration of a substance can have a topical effect when the administered molecule does not enter the circulatory system but only exerts its action locally, or a systemic effect, whereby the substance first enters the circulatory stream and is then distributed to the site of action. The main routes of administration with systemic effect are enteral and parenteral. The enteral route includes oral administration of the molecule. Although this route is one of the least invasive, it does not guarantee complete absorption of the administered substance. Parenteral routes include either administration through accessible mucous membranes or epithelia, or injection of the substance directly into the bloodstream. This route of administration is the preferred route for drugs that have poor absorption in the gastrointestinal tract (GIT) and drugs that are unstable in the GIT. As they bypass the GIT and are not absorbed by it, the bioavailability remains practically equal to the injected dose [18]. Parenteral administration has certain advantages, such as the avoidance of first-pass metabolism, better bioavailability, and reliable dosing. Compared to oral administration, parenteral administration enables control over dose and rate, thus generating more predictable pharmacodynamic and pharmacokinetic profiles [19].

Regarding cytokine studies until a few years ago, most studies in vivo have been conducted using the traditional parenteral route of administration [14]. However, parenteral administration can cause deposition of the administered cytokine in peripheral tissues, leading to some adverse events. In addition, because of the short half-life of cytokines, high doses are required to be effective, but this can lead to pleiotropic activity and adverse effects [20,21]. These side effects often include general mild and transient effects such as flu-like illness and injection site reaction, but more severe adverse reactions can also occur involving autoimmune, neuropsychiatric, ischemic and infection-related adverse events [22]. In view of the side effects of high doses of cytokines administered parenterally, it was necessary to evaluate other routes of administration that would allow the use of low doses of cytokines. Oral administration proved to be a viable alternative. When administered in low doses, cytokines exert their effects in a two-pronged mechanism of action. Initially, following the administration of cytokines, immunocompetent effector cells are activated in the oral lymphoid tissues and move towards the target site. Then, in the oral cavity, chemokines are produced, causing the redirection of the activated lymphocytes towards the infected cells in order to eliminate them [23]. Furthermore, scientific evidence shows that orally administered cytokines are able both to maintain their biological functions, despite the gastric and intestinal environment, and also to exert their function locally on the mucosa [24].

## 3. High vs. Low-Dose Cytokine Treatments

Ever since Isaacs and Lindenmann discovered interferons (IFNs) [25], the potential use of cytokines as therapeutic agents for homeostatic imbalances of the immune system has been perceived as a new and interesting field of research in biomedical science [26].

To date, different cytokines are used as therapeutic agents (Table 1) [27]. Cytokine treatments based on moderate concentrations after parenteral administration demand high dosage, since most cytokines used for clinical purposes have a short half-life [28]. However, due to the pleiotropic properties of cytokines, this type of administration also leads to the onset of different adverse effects, such as those mentioned above. In the last few decades, in order to overcome the problem of side effects, several studies have been carried out on the use of low-dose medicine (LDM), a therapeutic approach consisting in the use of cytokines and other biological molecules, administered at low dosages. A few studies have demonstrated the effectiveness of LDM in the treatment of both inflammatory diseases, such as allergic asthma [29], Crohn’s disease [30], vulgar psoriasis [31], and neoplasia [32,33]; accordingly, this new type of therapy seems to be a valid alternative to the use of high-dose cytokines [34]. In fact, the use of low-dose cytokines, with no or slight side effects, would be particularly useful in long-term treatments. At present, the limitation of this therapy is that patients with a compromised immune system require higher doses compared to those used in LDM, with the side effects that this entails [26].

## 4. Regional vs. Systemic Effects of Type I IFNs

Cytokines have been designed and formulated for diverse clinical purposes and different administration routes. The aforementioned adverse side effects of high-dose parenteral treatments have also been confirmed in veterinary medicine, and Type I IFNs are no exception to this general rule. Thus, the highest parenteral doses of IFN-α can trigger inflammatory and pyretic responses in cattle [58]. However, regional, anti-inflammatory control actions may co-exist with opposite systemic effects. This was the case of high-dose, parenteral, human interferon alpha 2 in patients with ulcerative colitis and Crohn’s disease [59]. Additionally, in human patients affected by Behcet’s disease (a systemic vasculitis), high-dose, parenteral IFN-α proved effective in reducing clinical signs, while retaining the adverse systemic side effects of high-dose treatment [60]. Likewise, beneficial effects of high-dose Type I IFNs following parenteral treatments were observed in veterinary medicine, too. In this respect, positive results have been demonstrated for a long time in the prophylaxis of diverse viral infections: Bovid herpesvirus I [61], bovine viral diarrhea virus [62], and vaccinia virus [63]. The therapeutic use of IFN-α also proved successful against enteric viruses, such as rotaviruses and coronaviruses [64]. Finally, parenteral yet much lower doses of human lymphoblastoid IFN-α (100–1000 IU/kg b.w.) proved effective in our experience as vaccine adjuvants [65] and immunomodulators for better thriftiness of cattle [66].

As for pets, high-dose parenteral treatment with human IFN-a2b proved effective against upper respiratory tract infections in shelter cats [67]. By far, the largest experience on high-dose IFN treatment was undertaken in cats and dogs with recombinant feline IFN-ω. In dogs, favorable results were reported in the treatment of Parvovirus infection [68,69] using 2.5 MU/kg over three days. Overall, recombinant feline IFN-ω has been licensed in several countries for treating canine parvovirus, feline leukemia virus, and feline immunodeficiency virus infections; it also exhibits a certain efficacy vis à vis other viral diseases [70]. Interestingly, in FIV cases, feline IFN-ω (0.1 MU daily) can be also administered as an oral therapy with significant clinical improvement in treated cats [71].

## 5. The Foundation of Oral, Low-Dose Cytokine Treatments

In the 70s and 80s, several scientific experiments demonstrated the existence of a bidirectional cross-talk, mediated by different signal molecules, between the psycho-neuroendocrine and immune systems [72]. This led to the development of the concept of psycho-neuro-endocrine-immunology (P.N.E.I.), according to which the biological functions of the body are interconnected, ensuring a homeostatic balance. The disruption of this equilibrium may lead to the development of diseases [73]. The homeostatic balance can be re-established through low-dose medicine (LDM), an approach based on the oral administration of biological molecules at concentrations close to physiological levels, in a range between 10^−9^ (nanograms/mL) and 10^−15^ M (femtograms/mL) [74]. Although the scientific literature demonstrated the effectiveness of oral cytokine therapy in modulating the immune response [75,76,77], one of the major difficulties of this type of therapy is the low bioavailability of the cytokines. This kind of constraint is likely to be overcome by oral, low-dose cytokine treatments supported by an effective drug delivery system (DDS) according to sequential kinetic activation (SKA) technology.

The SKA technology recalls two concepts at the basis of low-dose medicine: the dynamization of biological molecules and the memory of water. The process of the “dynamization” of biological molecules, whereby a substance is diluted and then shaken in a process called “succussion”, activates the “vital energy” of the substance that had been previously serially diluted. The water used for dilutions is affected by some of the “essential properties” of the substance that has been diluted as a result of shaking, in the conceptual framework of an outright “memory of the water” [78]. This term refers to the alleged property of water to retain a “memory” of the substances it has come into contact with. This controversial issue within the scientific community was first proposed by Jacques Benveniste, who in 1988 published an article in *Nature* suggesting that human basophil degranulation still works even in the absence of the inducing antibody [79]. Despite *Nature*’s denial of Benveniste’s findings [80], other scientists, including Nobel Prize Luc Montagnier, undertook experiments to demonstrate the validity of the assumptions made about the memory of water. In 2011, Luc Montagnier et al. published a study in the *Journal of Physics* showing how highly diluted aqueous solutions of DNA sequences from HIV, other viruses and bacteria can produce low-frequency electromagnetic signals characteristic of DNA in solution [81]. To date, although there have been studies that seem to confirm the “memory of water”, none of them have actually passed a double-blind trial; this may suggest a lack of repeatability for proper scientific recognition.

By exploiting the ability of the basic substance to release its pharmacological activity into the water environment [82], the SKA technology ensures that the biological molecules administered are active even at much lower concentrations (sub-nanomolar) than what is considered the minimum effective dose [74]. Therefore, the use of low-dose cytokines allows patients to be treated with results comparable to those induced by high concentrations, while avoiding the dose-dependent adverse effects.

## 6. Low-Dose IFN Treatments in Veterinary Species

To date, a wide variety of cytokines have been made available for the most important veterinary species. Cytokines in veterinary species have been investigated for the development of diagnostic tests, the study of protective immunity, the modulation of the immune response to vaccines and the development of preclinical models for human diseases [17]. The low-dose treatments referred to in the following chapters implied the use of 1 to 10 International Units (IU) IFN-α/kg b.w., which is 10^−3^ to 10^−5^ less compared to some parenteral cytokine treatments.

### 6.1. Livestock Animal Treatment

As livestock production has intensified, the addition of antibiotics and chemical antimicrobials to animal feed in an attempt to prevent disease has become a common practice. This procedure, however, has proved to be a double-edged sword: while it has beneficial effects, it also leads to an increased prevalence of antibiotic-resistant bacteria [83]. In the face of this evidence, the WHO has recommended the use of alternative methods to control diseases, and among these, the use of cytokines has proven to be a viable alternative to conventional antibiotic therapies [84].

#### 6.1.1. Ruminants

In ruminants, and particularly in cattle, the use of orally administered interferon-α (IFN-α) has been shown to be safe and effective in the treatment of several diseases [85]. Positive results were mostly obtained in young cattle with no or limited ruminal functions, whereas this was not true of adult dairy cows in the pregnancy to lactation transition period [86]. In cattle with bovine respiratory disease (BRD), a single dose of natural HuIFN-α has been shown to reduce morbidity and mortality associated with BRD in farmed cattle [87]. Additionally, in cattle, the oral administration of natural HuIFN-α proved useful in controlling infection with *Theileria parva* experimental infection, providing protection against East Coast fever (ECF) [88]. A study conducted in Japan also proved the efficacy of low-dose natural HuIFN-α for the oral treatment of cattle suffering from rotavirus diarrhea [89]. The addition of recombinant HuIFN-α to the milk replacer used to feed calves proved beneficial for them. Indeed, this intervention reduced the severity of diarrhea and the incidence of otitis in calves fed milk replacer supplemented with recombinant HuIFN-α [90]. In bulls and steers infected with infectious bovine rhinotracheitis (IBR) virus, a treatment with low-dose natural HuIFN-α proved effective in terms of fever decrease, with a consequent reduction in the use of antibiotics, and an increase in body weight [91]. In 2006, Namangala and colleagues proved that the oral administration of low-dose natural HuIFN-α in cattle could be a potent immunostimulant. In fact, IFN-stimulated genes (ISGs), in particular those involved in key biological functions, are upregulated following treatment with IFN-α. This is the case of lysozyme C-2 precursor, chemokine orphan receptor 1, immediate early response 5, phagocytic glycoprotein 1 (CD44 antigen precursor) and chromatin licensing and DNA replication factor 1 (CDT1), to cite just the five most upregulated ones [92]. According to this result, oral IFN-α treatments can be used as prophylactic tools against some viral and neoplastic conditions because of the increased response of the host’s immune system [92].

#### 6.1.2. Pig

Since the 1990s, different publications have demonstrated the efficacy of oral, low-dose IFN treatment in pigs. In neonate piglets infected with porcine enteric rotavirus or transmissible gastroenteritis virus (TGEV), the oromucosal administration of natural HuIFN-α or its addition in a liquid diet was shown to lead to a significant benefit in terms of weight gain. The mechanisms by which HuIFN-α administration is able to provide benefits in TGEV-infected piglets is not entirely clear [93,94]. However, it is known that porcine blood mononuclear cells secrete IFN-α after induction with TGEV, and that interferon production increases proportionally with age [95]. In 2002, due to the emerging problem of PRRS (porcine reproductive and respiratory syndrome) and PMWS/PDNS (post-weaning multisystemic wasting syndrome/porcine dermatitis and nephropathy syndrome), Amadori and colleagues conducted a study to evaluate the efficacy of the oral administration of IFN-α in piglets to control PRSS and PMWS. The results of this study firstly demonstrated that IFN-α-treated animals showed an improvement in both the clinical course of the disease and weight gain in “problem” herds with high losses around weaning, while herds with lower losses presented a significant improvement in weight gain only. Finally, treatment with IFN-α led to a significant reduction in the drugs routinely used in the herds [96]. This study was based on the evidence that certain viral agents, such as PRRSV and porcine circovirus 2, could act to increase the sensitivity of alveolar macrophages [97]. Therefore, stimulating an adequate homeostatic response may be useful in reducing the release of inflammatory cytokines in the lungs [98]. Another study by Amadori and colleagues has shown that inflammatory cytokines play an important role in the environmental adaptation strategies of farm animals. In particular, an in vivo model of early weaning was used to assess the contribution of IFN-α to homeostatic functions. This study first of all confirm the role of type I IFNs in the control of the inflammatory response at weaning, showing that orally administered, low-dose IFN-α can exert a control action on the inflammatory cytokine response under adverse environmental conditions, similar to that exerted by endogenous IFN-α in nature. This treatment may therefore decrease the need for the host’s own response to the weaning stress, although further studies are necessary to fully understand this phenomenon [99]. More recently, Fan and colleagues evaluated the antiviral effect of using recombinant porcine IFNs (PoIFN-α and PoIFN-γ) as an emergency treatment for pigs infected by African swine fever virus (ASFV) [100]. Their results confirmed previous data [101,102], demonstrating that the administration of low doses of recombinant porcine IFN (10^5^ IU/kg) can significantly increase cytokine expression, significantly reduce viral replication, and alleviate clinical signs during early infection [100].

#### 6.1.3. Poultry

Several studies on the efficacy of oral IFN administration have also been conducted in chickens, partly because the use of cytokines in this species became more feasible with the recent cloning of different avian cytokines [84]. With regard to viral infections, a study by Marcus et al. observed that the preventive oral administration of recombinant chicken (Ch)IFN-α via drinking water can provide protection against Newcastle disease virus (NDV) infection, whereas the treatment of infected animals delayed the onset and severity of clinical symptoms, as well as preventing histopathological lesions in the trachea [103]. In addition, the oral administration of recombinant ChIFN-α via drinking water was also shown to significantly reduce the replication of Marek’s disease virus [104]. More recently, Meng et al., in a study on SPF chickens experimentally infected with H9N2 avian influenza virus (AIV), demonstrated that oral administration of ChIFN-α not only ensured the rapid recovery of body weight gain in infected chickens, but also exerted an antiviral effect. This study therefore suggested that ChIFN-α could be used as both a preventive and a therapeutic agent against AIV [105]. This study showed that ChIFN-α treated chickens, following infection with AIV H9N2, increased their expression of both 2′,5′- oligoadenylate synthetase (2′,5′-OAS) and myxovirus resistance proteins 1 (Mx1) compared to the control ones. Therefore, it is believed that 2′,5′-OAS and Mx1, at least in part, are involved in the antiviral effector functions of ChIFN-α. Previous studies [106,107,108], however, had shown that the Mx1 protein did not possess antiviral activity, so the expression level of Mx1 and 2′,5′-OAS observed in this study is not sufficient to explain the antiviral mechanism of ChIFN-α, thus indicating that other antiviral ISGs, such as protein kinase (PKR) and adenosine deaminase RNA (ADAR1), might be involved in the antiviral mechanism of ChIFN-α. In support of Meng et al.’s results, an in vitro experiment showed that the pre-treatment of chicken and turkey lung cells with 1000 IU/mL rChIFN-α, prior to infection with H1N1 and H5N9 viruses, not only dramatically reduced viral replication, but also decreased both interferon and proinflammatory responses [109]. This study shows that rChIFN-α leads to a reduction in viral titres and a decrease in viral nuclear protein (NP) production, thereby limiting AIV replication. Since NP plays an important role in the structural maintenance of the ribonucleoprotein complex, as well as in genome replication, by interacting with viral RNA [110,111], its reduction could be one of the mechanisms by which rCHIFN-α treatment exerts an antiviral effect [109]. The oral administration of cytokines in chickens was shown to be effective not only in the treatment of viral diseases, but also in alleviating the weight loss due to environmental stress factors, as demonstrated in a study by Fulton et al. [112].

### 6.2. Pet Animals

#### 6.2.1. Dog

The oral administration of low-dose IFN-α in dogs has proven useful for the treatment of several diseases. In dogs with keratoconjunctivitis sicca, which are mainly treated with surgical correction or through the use of artificial tears, oral administration of HuIFN-α has been shown to lead to increased tearing, making it an effective therapeutic alternative for the treatment of keratoconjunctivitis sicca [113]. The effects of IFN-α in dogs were very similar to those observed in humans treated for Sjögren’s syndrome. Indeed, since 1993, based on the experience of Shiozawa and colleagues [114], oral treatment with recombinant HuIFN-α in patients with Sjögren’s syndrome has been shown to be effective in terms of salivary gland function, as well as the histological improvement of minor salivary gland lesions [115]. Another study demonstrated the efficacy of HuIFN-α oral treatment in dogs affected by idiopathic recurrent superficial pyoderma in terms of amelioration of the mean clinical scores and a decreased need for antimicrobials [116]. In addition, a study by Stokking and colleagues demonstrated that oral administration of HuIFN-α also leads to an improvement in the clinical condition of dogs suffering from pigmented epidermal plaques [117]. More recently, other studies have evaluated the efficacy of the oral administration of non-human IFNs in dogs with different pathological conditions. Ito et al. conducted a study to evaluate the therapeutic efficacy of the low-dose oral administration (LDOA) of recombinant canine IFN-α subtype 4 (CaIFN-α4) in dogs with periodontal disease. They found that the administration of CaIFN-α4 reduced periodontopathic bacterial counts, stress responses, and gingival inflammation. Hence, the use of LDOA of CaIFN-α4 may be considered as a new possibility to treat dogs affected by periodontal disease [118]. The mechanism of how orally administered IFN-α affects the systemic/local immune system is still unclear, but is suggested that IFNs are able to up-regulate IL-8 production by gingival fibroblasts in response to bacterial and cytokine components, thereby modulating the inflammatory response in periodontal tissues [119]. Additionally, Litzlbauer and colleagues reported that the oral administration of 500 to 2000 IU/kg b.w. recombinant feline interferon omega (rFeIFN-ω) was effective in treating dogs with canine atopic dermatitis (CAD). Their study showed that oral treatment led to greater improvements compared to subcutaneous treatment with the same IFN molecule. The improvements, assessed through the Canine Atopic Dermatitis Extent and Severity Index (CADESI), mainly concerned lesions, while in both IFN dose groups there were no significant changes in pruritus or quality of life, these two parameters being evaluated by the animals’ owners [120].

#### 6.2.2. Cat

The oral administration of low-dose HuIFN-α was also undertaken in cats with feline leukemia virus (FeLV) infection. FeLV cats receiving the treatment survived longer than control cats following experimental infection with FeLV. In addition to that, only a small percentage of treated cats developed clinical disease during the study, whereas all control cats did, despite a similar viral load [15,121]. In a work conducted by Pedretti et al., cats infected by feline immunodeficiency virus (FIV) and showing overt AIDS-related disease symptoms were treated with low-dose oral human interferon-α or placebo [16]. This work described that after HuIFN-α administration, the survival of infected cats was significantly increased, with a rapid and dramatic improvement in the clinical score in infected hosts. This study also found that although HuIFN-α therapy failed to maintain the balance between CD4+ and CD8+ T lymphocytes, there was good survival of CD4+ T cells and a gradual increase in CD8+ T cells. In addition, a strong correlation was noted between the leukocyte count and the condition of the cats tested. Indeed, the highest leukocyte counts were observed in HuIFN-α-treated cats, indicating a strengthened innate immune response and better control of opportunistic infections (Figure 1) [16]. Similar observations were presented in the work of Gomez-Lucia et al., where naturally FeLV and FIV infected cats orally treated with low IFN-α doses (recombinant HuIFN-α, 60 IU/day) ameliorated their clinical conditions, including reduced proviral load, reduction of the anemia, a more favorable CD4+/CD8+ cell ratio, and increased concentrations of gammaglobulins. They also observed that the therapeutical protocol should be prolonged or at least organized in cycles [122,123]. In other studies, the efficacy of rHuIFN-α in the therapy of cats affected by FeLV was not confirmed [124]. The oral use of feline IFN-ω has not been deeply investigated. For example, in a study without the placebo control, the viral load did not change, but an improvement in the general clinical conditions was observed after the treatment of cats affected by FIV. This was related to a major effect on secondary infection, and not a direct effect on FIV itself [125,126]. Finally, a prospective randomized placebo-controlled clinical trial demonstrated a significant improvement in clinical signs and viral load of unvaccinated FHV-1-positive cats treated orally with low-dose IL-12 plus IFN-γ (10 fentograms/mL twice daily for 6 months) [127].

#### 6.2.3. Horse

Two studies by Moore BR et al. and Moore I et al. [128,129] described the oral administration of IFN-α in racehorses. The treatment appeared to be useful in restoring the natural homeostatic mechanisms in respiratory tract tissues of racehorses suffering from inflammatory airway diseases [14]. In these studies, oromucosal administration of natural HuIFN-α was able to alleviate the clinical signs of inflammatory airway disease (IAD) [104], stopping cough after 4 weeks of treatment [128]. The mechanisms by which IFN-α is able to produce a therapeutic effect in IAD-affected horses have not been established with certainty; it is thought that IFN-α can exert an antiviral control action by inducing the production of enzymes that inhibit the synthesis of viral proteins and degrade viral RNA [129]. More recently, Scudo and colleagues conducted a further study on the prevention of respiratory diseases in the English thoroughbred galloper. They observed the efficacy of the homeopathic drug named Transfactor 21, containing lymphoblastoid HuIFN-α, in two groups of subjects (Group 1: 16–20-month-old foals that had just arrived on the stud farm. Group 2: 24–30-month-old horses undergoing daily training outings). In both groups the animals treated with Transfactor 21 fell ill with a lower incidence, less severe symptoms, and shorter duration of illness than the untreated animals [130]. The possible use of orally administered IFN-α has also been considered to prevent the occurrence of transport fever in thoroughbred racehorses [131,132]. In fact, in the long-distance transfer of these animals, episodes of pneumonia with pyrexia often occur due to the accumulated transportation stress [133]. Studies have shown that, although IFN-α could not prevent the onset of carriage fever, it was useful in reducing symptoms. In fact, animals treated with IFN-α, when compared to the control group, presented lower values of rectal temperatures, white blood cell counts, plasma fibrinogen concentrations and serum amyloid A concentrations immediately after transport [131,132]. Additionally, horses given IFN-α three times prior to transport had a lower incidence of disease than both the horses treated once and the untreated ones [132]. These results show that, once the correct dosage and number of doses have been identified, the low-dose administration of IFN-α could prevent the onset of carriage fever and/or its serious clinical outcome [131].

### 6.3. Laboratory Animals

Evaluation of the effectiveness of oral IFN administration has been carried out mainly in rodents. Studies have evaluated the use of IFNs in the treatment of various infectious, autoimmune, or experimentally induced neoplastic diseases. With regard to infectious diseases, the efficacy of oral IFN treatment has proven useful in the treatment of several viral infections. Several studies have evaluated the efficacy of the oral administration of murine IFN-α (MuIFN) and MuIFN-β. Among these, Schafer et al. showed that the addition of these interferons to milk resulted in a decreased mortality rate in neonatal mice experimentally infected with vesicular stomatitis virus [134]. Nagao et al. studied the efficacy of the oral administration of MuIFN-α in mice challenged with vaccinia virus (VV). Following the low-dose administration of MuIFN-α, pock formation on the tail skin of infected mice was significantly suppressed. This demonstrated that the oral administration of IFN-α at low doses is able to activate an immune response and thus protect mice from VV infection. This study did not lead to a clear definition of the mechanism of action of oralmucosally administered recombinant murine IFN-α, but the authors hypothesized that it may act on perioral lymphoid tissue in order to activate lymphoid cells in situ, and is then transported to peripheral immunologic organs via efferent lymphatic vessels to enhance the immune response [135]. Additionally, Tovey and colleagues confirmed that the administration of 10^4^ IU of rMuIFN-α/β was able to ensure increased survival in mice challenged with a lethal dose of vesicular stomatitis virus (VSV) [136]. In addition, the oral administration of MuIFN-α/β (from 2 to 10^5^ IU) in encephalomyocarditis virus (EMCV)-infected mice was also shown to increase the survival rate of the animals in a dose-dependent manner [136]. Finally, Tovey and colleagues noticed that in BALB/c mice, previously infected with an intranasal administration of varicella zoster virus (VZV), the oromucosal administration of MuIFN-α e MuIFN-β inhibited virus replication in the spleen, lungs and brain [136]. The results of the study by Tovey and colleagues show that the mechanism of action of oromucosal IFN-α therapy, although not fully identified, may be different from that of parenterally administered IFN-α. It was found that the oral administration of IFN-α is ineffective prior to viral infection as opposed to parenteral administration [136]. Additionally, for murine cytomegalovirus, the replication was inhibited in the spleen and liver of BALB/c after oromucosal administration of recombinant MuIFN-α or MuIFN-α in combination with MuIFN-β [137,138,139]. More recently, it has been shown that the oral administration of IFN-α expressed in *Bifidobacterium longum* significantly reduced the severity of the myocarditis in BALB/c mice infected with coxsackie virus B3 (CVB3), reducing the virus titres in the heart and inducing a Th1 pattern of response in the spleen and heart, with a consequent reduction in the inflamed cardiac area [140].

Other studies have shown that in mice with experimentally induced neoplastic diseases, oral IFN-α administration can induce beneficial effects. In fact, the administration of MuIFN-α and MuIFN-β (10^5^ units) in DBA/2 mice challenged with Friend leukemia cells (FLCs) led to an increased survival rate compared to the control mice [136,141]. Additionally, by administering HuIFN-α in the drinking water, an antitumor response was induced in C57Bl/6 mice challenged with B16 melanoma cells [142].

Transgenic mice have been used to study certain human autoimmune diseases. Segerer and colleagues have used mixed cryoglobulinemia, thymic stromal lymphopoietin-deficient transgenic mice in order to mimic the human mixed cryoglobulinemia disease. In this study, the administration of universal type I IFN induced the amelioration of three parameters: mean glomerular tuft area, mean glomerular areas occupied by macrophages, and mean number of inflammatory cells per glomerulus compared with the control animals [143]. A study by Gariboldi et al. assessing the therapeutic efficacy of cytokine administration in Th2-type disease showed that, in a murine model of allergic asthma, very low dosages of dynamized solutions of IL-12 and IFN-γ, co-delivered by oral route, are able to revert the pathologic condition of the mice, restoring the normal balance between Th1 and Th2 cytokines, thus bringing mice to a healthy state [29].

## 7. Other Approaches

Cytokines and IFNs are not the only biological molecules that can be used therapeutically. Since Lawrence’s hypothesis of their existence in the early 1950s [144], studies have focused on transfer factors (TFs), low-molecular weight molecules extracted from lymphocytes that are able to transfer antigen-specific information for cell-mediated immunity (CMI) from an immunized donor to a naïve recipient. Studies of these molecules have shown their potential therapeutic use against viral, parasitic, fungal and mycobacterial infections [145].

## 8. Conclusions

Over the years, cytokine research has led to the study and development of new therapies that have revolutionized the treatment of certain diseases, such as cancer, autoimmune diseases, and certain types of infectious diseases. Several studies have shown that cytokine administration plays an important regulatory role in the inflammatory response. The use of cytokines has proven useful not only in the field of human medicine, but has also found use in the veterinary field. In particular, animal studies have evaluated the efficacy of low-dose oral IFN-α treatment in many animal species, such as rodents, ruminants, pigs, horses, dogs and cats, in order to treat viral infections and immune-mediated disorders. Although the use of IFNs has proven useful and effective in the treatment of farm animals, it cannot currently be advocated as an alternative to antibiotics because of legal constraints. This issue is of paramount importance. Due to the intensification of animal production, the amount of antibiotics used to combat opportunistic diseases has increased significantly, resulting in a deterioration of food safety and an increase in the phenomenon of antibiotic resistance. To cope with this, the use of alternative remedies, such as cytokines, has been recommended. Unfortunately, however, due to the current legal framework regulating the use of drugs in livestock, useful treatments such as low-dose IFN-α or IL-2 cannot be prescribed to farm animals, because the active ingredients of admitted drugs must be included in an old list of authorized molecules with or without maximum residue levels (MRLs), listed in specific annexes of COUNCIL REGULATION (EEC) No. 2377/90. Moreover, investigations into effective protocols based on the administration of low-dose cytokines were abandoned late in the 1990s by big pharma in favor of more profitable drugs, such as antibiotics. In contrast, the use of cytokines is permitted in companion animals; thus, IFN-α is widely used for the long-term treatment of FeLV- and FIV-infected cats. Thus, although IFN-α has been officially approved by EFSA and the EMA, as a useful alternative to antibiotics (https://www.ema.europa.eu/en/events/committee-medicinal-products-veterinary-use-cvmp-6–8-december-2016; accessed on 7 March 2021), its use, like that of other immunomodulators, remains banned in farm animals, despite the actual need.

Alternative therapies based on cytokines and transfer factors have several advantages, such as efficacy and reduced toxicity. Therefore, they deserve to be investigated further in order to clarify the precise mechanisms of action and identify the correct administration schedules.

## Figures and Tables

**Figure 1 vetsci-08-00109-f001:**
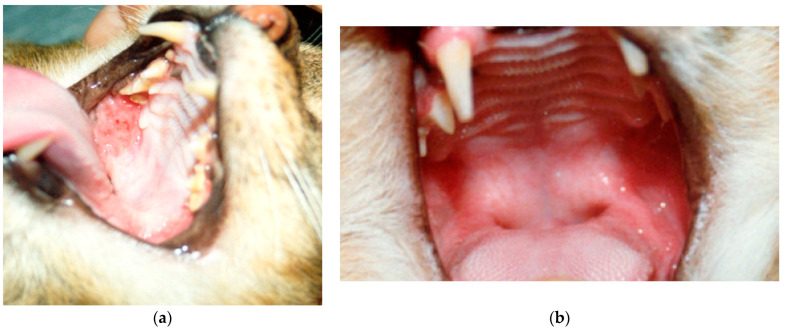
Effect of IFNα treatment in cats affected by FIV. (**a**) Opportunistic lesions in the oral cavity of a cat affected by FIV, before IFN oral treatment; (**b**) resolution of the oral cavity lesions after IFN oral treatment in the same cat [16].

**Table 1 vetsci-08-00109-t001:** Cytokines used and approved as therapeutic agents in human medicine.

Cytokines	Disease’s Target	References
IFN-α	Hepatitis B; hepatitis C	[35,36]
IFN-β	Multiple sclerosis	[37]
IFN-γ	Chronic granulomatous disease; Crohn’s disease	[38,39]
TNF-α	Rheumatoid arthritis, psoriasis; Crohn’s disease; ankylosing spondylitis; chronic obstructive pulmonary disease; sepsis; juvenile idiopathic arthritis; asthma	[40,41,42,43]
G-CSF	Febrile neutropenia; bone marrow transplantation	[44]
GM-CSF	Neutropenia after chemotherapy	[45]
IL-1	Rheumatoid arthritis; juvenile idiopathic arthritis; Still’s disease; Crohn’s disease	[42,46]
IL-2	Metastatic renal cell carcinoma; renal transplantation	[47,48,49]
IL-4	Asthma	[50]
IL-5	Asthma	[51]
IL-6	Rheumatoid arthritis	[52,53]
IL-8	Melanoma	[54]
IL-10	Crohn’s disease; rheumatoid arthritis; psoriasis; ulcerative colitis; multiple sclerosis	[55]
IL-11	Thrombocytopenia; ulcerative colitis; psoriasis; rheumatoid arthritis; Crohn’s disease	[56]
EPO	Anemia	[57]

IFN = interferon; TNF-α = tumor necrosis factor-alpha; G-CSF = granulocyte colony stimulating factor; GM-CSF = granulocyte-macrophage colony stimulating factor; IL = interleukin; EPO = erythropoietin.

## Data Availability

Not applicable.

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
