# Peer review of "Therapeutic and Prophylactic Use of Oral, Low-Dose IFNs in Species of Veterinary Interest: Back to the Future"

_vetsci, 2021, doi:10.3390/vetsci8060109_

Round 1
Reviewer 1 Report
The manuscript by Frazzini and coworkers on the therapeutic and prophylactic us of oral low dose cytokines in animals is interesting and well-written. After a description of cytokines, what they are and their use in human medicine, the authors review some applications of low-dose interferon in veterinary medicine. I recommend its publication. However, I would like the authors to consider some observations.
The manuscript emphasizes constantly “oral-use of cytokines” when in fact only IFN studies are referred. Please, review thoroughly and change cytokines to interferon, even in the title, so that the reader is not expecting results of, for example, IL-2, which is also used in dogs for treating neoplasias.
Though the mechanism of action is well-explained and probably does not deviate widely from the present knowledge, more recent references should sustain how oral-administered cytokines (specifically IFN) act.
The main body of the manuscript, the oral administration of low-dose IFN does not relate to a homeopathic administration, as implied in heading 4, as this would use even a lower dose. Cytokines are known to act as low doses, and it has been shown that they are absorbed in the oronasal tissues when administered orally, where the concentration is not as low as if they were administered at homeopathic concentrations. Thus, this section is misleading and should be removed, at least L137-162.
Please, check carefully that it is specified whether the molecules used are recombinant or natural; natural substances, unless very well purified, may include some other substances which mask results.
Gomez-Lucia et al (2019 and 2020) have also used orally administered low-dose rHuIFN to treat FeLV- and FIV-infected cats, and their results should also be discussed (doi:10.3390/v11090845 and doi:10.3390/ani10091464)
A paragraph on studies that show that oral low doses of interferon are not effective should also be included. For example, Hartmann has reviewed the different treatments for FeLV and FIV and does not favor the oral administration.
Please, rephrase the conclusions (L428-434) as it is repetitive and difficult to follow
Very minor observations.
L31 Please, change “can break down due an” to “can break down due to an”.
L59 Please, change FELV to FeLV, as it is used later in the manuscript.
L90 Please change route to routes.
Table 1. Please, complete the legend indicating that it refers exclusively to human medicine, and whether treatments are experimental or approved for routing use. In addition, please, cite correctly and remove name initials from the references.
L 165 As all the discussion deals with IFN, change the heading accordingly.
L171, 370 and 373 (at least) Check whether the IFN units are correct.
L178 Please, cite correctly.
L278, 311, 315 and others Please, be consistent in the way IFN is cited. In addition, in several lines the type of IFN is not shown. Or in places, such as L340, the species where IFN was obtained from is not indicated.
L327 Please, add the abbreviation IAD after the first use of the words.
References Please, check them as some have the names in capitals and some not, in some the title is in capitals, etc.
Author Response
Revision note
Manuscript: vetsci-1216451
Dear Editor,
we revised our manuscript according to the reviewers’ comments.
Changes were highlighted using underlined red text in the present revised version of our manuscript.
Reviewer 1
The manuscript by Frazzini and coworkers on the therapeutic and prophylactic us of oral low dose cytokines in animals is interesting and well-written. After a description of cytokines, what they are and their use in human medicine, the authors review some applications of low-dose interferon in veterinary medicine. I recommend its publication. However, I would like the authors to consider some observations.
The manuscript emphasizes constantly “oral-use of cytokines” when in fact only IFN studies are referred. Please, review thoroughly and change cytokines to interferon, even in the title, so that the reader is not expecting results of, for example, IL-2, which is also used in dogs for treating neoplasias.
Thank you for your suggestion, we replaced “cytokine” with “IFN” in the majority of the cases all over the manuscript.
Though the mechanism of action is well-explained and probably does not deviate widely from the present knowledge, more recent references should sustain how oral-administered cytokines (specifically IFN) act.
Thank you for your suggestion, we added the following references:
- Hilton, L.S.; Bean, A.G.D.; Lowenthal, J.W. The emerging role of avian cytokines as immunotherapeutics and vaccine adjuvants. Vet. Immunol. Immunopathol. 2002, 85, 119–128, doi:10.1016/S0165-2427(01)00414-7.
- Mamber, S.W.; Lins, J.; Gurel, V.; Hutcheson, D.P.; Pinedo, P.; Bechtol, D.; Krakowka, S.; Fields-Henderson, R.; Cummins, J.M. Low-dose oral interferon modulates expression of inflammatory and autoimmune genes in cattle. Vet. Immunol. Immunopathol. 2016, 172, 64–71, doi:10.1016/j.vetimm.2016.03.006.
- Fan, W.; Jiao, P.; Zhang, H.; Chen, T.; Zhou, X.; Qi, Y.; Sun, L.; Shang, Y.; Zhu, H.; Hu, R.; et al. Inhibition of African Swine Fever Virus Replication by Porcine Type I and Type II Interferons. Front. Microbiol. 2020, 11, doi:10.3389/fmicb.2020.01203.
The main body of the manuscript, the oral administration of low-dose IFN does not relate to a homeopathic administration, as implied in heading 4, as this would use even a lower dose. Cytokines are known to act as low doses, and it has been shown that they are absorbed in the oronasal tissues when administered orally, where the concentration is not as low as if they were administered at homeopathic concentrations. Thus, this section is misleading and should be removed, at least L137-162.
Thank you for your suggestion, we deleted the mentions to homeopathy, but we let the description of the SKA technology (lines 140-144) because we think it is an important concept also reported in some references.
We re-phrased the paragraph as follows: “The SKA technology recalls two concepts at the basis of low-dose medicine: the dynamization of biological molecules and the memory of water. The process of “dy-namization” of biological molecules, whereby a substance is diluted and then shaken in a process called “succession”, activating the “vital energy” of the substance that had been previously serially diluted. The water used for dilutions is affected by some of the 'es-sential properties' of the substance that has been diluted as a result of shaking, in the conceptual framework of an outright “memory of the water” [78]”.
Please, check carefully that it is specified whether the molecules used are recombinant or natural; natural substances, unless very well purified, may include some other substances which mask results.
Thank you for your suggestion, we added in the text when the molecules were recombinant or natural all over the manuscript.
Gomez-Lucia et al (2019 and 2020) have also used orally administered low-dose rHuIFN to treat FeLV- and FIV-infected cats, and their results should also be discussed (doi:10.3390/v11090845 and doi:10.3390/ani10091464).
Thank you for your suggestion, we discussed the results of Gomez-Lucia and colleagues (lines 358-365) and we added the related references.
- Gomez-Lucia, E.; Collado, V.M.; Miró, G.; Martín, S.; Benítez, L.; Doménech, A. Follow-up of viral parameters in felv-or Fiv-naturally infected cats treated orally with low doses of human interferon alpha. Viruses 2019, 11, doi:10.3390/v11090845.
- Gomez-Lucia, E.; Collado, V.M.; Miró, G.; Martín, S.; Benítez, L.; Doménech, A. Clinical and hematological follow-up of long-term oral therapy with type-i interferon in cats naturally infected with feline leukemia virus or feline immunodeficiency virus. Animals 2020, 10, 1–14, doi:10.3390/ani10091464.
A paragraph on studies that show that oral low doses of interferon are not effective should also be included. For example, Hartmann has reviewed the different treatments for FeLV and FIV and does not favor the oral administration.
Thank you for your suggestion, we discussed also about study that reported any effect of IFN in the treatment of FIV and FeLV (lines 164-165) and added the related references.
- McCaw, D.; Boon, G.; Jergens, A.; Kern, M.; Bowles, M.; Johnson, J. Immunomodulation therapy for feline leukemia virus infection. J. Am. Anim. Hosp. Assoc. 2001, 37, 356–363, doi:10.5326/15473317-37-4-356.
Please, rephrase the conclusions (L428-434) as it is repetitive and difficult to follow.
Thank you for your suggestion, we re-phrased the sentence as follows (lines 477-481): “In particular, animal studies have evaluated the efficacy of low-dose, oral IFN-a treatment in many animal species, such as rodents, ruminants, pigs, horses, dogs and cats, in order to treat viral infections and immuno-mediated disorders”.
Very minor observations.
L31 Please, change “can break down due an” to “can break down due to an”.
Thank you for your suggestion, we corrected the sentence.
L59 Please, change FELV to FeLV, as it is used later in the manuscript.
Thank you for your suggestion, we corrected the sentence.
L90 Please change route to routes.
Thank you for your suggestion, we corrected the sentence.
Table 1. Please, complete the legend indicating that it refers exclusively to human medicine, and whether treatments are experimental or approved for routing use. In addition, please, cite correctly and remove name initials from the references.
Thank you for your suggestion, we completed the legend by adding that the cytokines mentioned are used and approved in human medicine.
L 165 As all the discussion deals with IFN, change the heading accordingly.
Thank you for your suggestion, we corrected the heading.
L171, 370 and 373 (at least) Check whether the IFN units are correct.
Thank you for your suggestion, we corrected the IFN units.
L178 Please, cite correctly.
Thank you for your suggestion, we corrected the reference.
L278, 311, 315 and others Please, be consistent in the way IFN is cited. In addition, in several lines the type of IFN is not shown. Or in places, such as L340, the species where IFN was obtained from is not indicated.
Thank you for your suggestion, we mentioned IFN in the same manner all over the text and we added the information about the species of origin of the IFN.
L327 Please, add the abbreviation IAD after the first use of the words.
“Inflammatory airway disease (IAD)” is mentioned for the first time at Line 384; later in the text only the abbreviation is used (line 386 “IAD-affected horses”).
References Please, check them as some have the names in capitals and some not, in some the title is in capitals, etc.
Thank you for your suggestion, we corrected the references.
Reviewer 2 Report
This review describes the use of oral cytokine therapy for veterinary applications. The review is organized and easy to follow and presents some interesting information. However, more detail and a more exhaustive literature search is needed for each of the sections. For example, the areas of canine cytokine therapy are brief and cursory. This review could easily double the information for those two sections, presumably this holds true for the other species in the review (chickens, cattle, horses etc). The authors must be more extensive in their literature search for this review to be acceptable for publication. Perhaps they should also mention non-oral cytokines for the purposes of completeness and whether these might be formulated for oral use, for example feline IFN omega.
Minor Issues:
Line 50-53: this sentence is unclear and probably considered a "run on" sentence
Lines 199-200: Which IFN genes are stimulated? Be specific
The word "showed" is used repeatedly, better choices would be "reported, observed, demonstrated"
Author Response
Revision note
Manuscript: vetsci-1216451
Dear Editor,
we revised our manuscript according to the reviewers’ comments.
Changes were highlighted using underlined red text in the present revised version of our manuscript.
Reviewer 2
This review describes the use of oral cytokine therapy for veterinary applications. The review is organized and easy to follow and presents some interesting information. However, more detail and a more exhaustive literature search is needed for each of the sections. For example, the areas of canine cytokine therapy are brief and cursory.
Thank you for your suggestion, we deepen the literature search restricted mainly on oral use of IFN low dose and we added the discussion in the relative sections.
We added the following references:
- Hilton, L.S.; Bean, A.G.D.; Lowenthal, J.W. The emerging role of avian cytokines as immunotherapeutics and vaccine adjuvants. Vet. Immunol. Immunopathol. 2002, 85, 119–128, doi:10.1016/S0165-2427(01)00414-7.
- Mamber, S.W.; Lins, J.; Gurel, V.; Hutcheson, D.P.; Pinedo, P.; Bechtol, D.; Krakowka, S.; Fields-Henderson, R.; Cummins, J.M. Low-dose oral interferon modulates expression of inflammatory and autoimmune genes in cattle. Vet. Immunol. Immunopathol. 2016, 172, 64–71, doi:10.1016/j.vetimm.2016.03.006.
- Fan, W.; Jiao, P.; Zhang, H.; Chen, T.; Zhou, X.; Qi, Y.; Sun, L.; Shang, Y.; Zhu, H.; Hu, R.; et al. Inhibition of African Swine Fever Virus Replication by Porcine Type I and Type II Interferons. Front. Microbiol. 2020, 11, doi:10.3389/fmicb.2020.01203.
- Esparza, I.; Gonzalez, J.C.; Vinuela, E. Effect of Interferon- , Interferon- and Tumour Necrosis Factor on African Swine Fever Virus Replication in Porcine Monocytes and Macrophages. J. Gen. Virol. 1988, 69, 2973–2980, doi:10.1099/0022-1317-69-12-2973.
- Paez, E.; Garcia, F.; Gil Fernandez, C. Interferon cures cells lytically and persistently infected with African swine fever virus in vitro. Arch. Virol. 1990, 112, 115–127, doi:10.1007/BF01348989.
- Weiss RC, C.J. and R.A. Low-dose orally administered alpha interferon treatment for feline leukemia virus infection. J Am Vet Med Assoc 1991, 199, 1477–81.
- Gomez-Lucia, E.; Collado, V.M.; Miró, G.; Martín, S.; Benítez, L.; Doménech, A. Follow-up of viral parameters in felv-or Fiv-naturally infected cats treated orally with low doses of human interferon alpha. Viruses 2019, 11, doi:10.3390/v11090845.
- Gomez-Lucia, E.; Collado, V.M.; Miró, G.; Martín, S.; Benítez, L.; Doménech, A. Clinical and hematological follow-up of long-term oral therapy with type-i interferon in cats naturally infected with feline leukemia virus or feline immunodeficiency virus. Animals 2020, 10, 1–14, doi:10.3390/ani10091464.
- McCaw, D.; Boon, G.; Jergens, A.; Kern, M.; Bowles, M.; Johnson, J. Immunomodulation therapy for feline leukemia virus infection. J. Am. Anim. Hosp. Assoc. 2001, 37, 356–363, doi:10.5326/15473317-37-4-356.
- Doménech, A.; Miró, G.; Collado, V.M.; Ballesteros, N.; Sanjosé, L.; Escolar, E.; Martin, S.; Gomez-lucia, E. Veterinary Immunology and Immunopathology Use of recombinant interferon omega in feline retrovirosis : From theory to practice. 2011, 143, 301–306, doi:10.1016/j.vetimm.2011.06.008.
- Hartmann, K. Efficacy of antiviral chemotherapy for retrovirus-infected cats . What does the current literature tell us? 2015, 925–939, doi:10.1177/1098612X15610676.
- Fiorito, F.; Cantiello, A.; Elvira, G.; Navas, L.; Diffidenti, C.; Martino, L. De; Maharajan, V.; Olivieri, F.; Pagnini, U.; Iovane, G. Comparative Immunology , Microbiology and Infectious Diseases Clinical improvement in feline herpesvirus 1 infected cats by oral low dose of interleukin-12 plus interferon-gamma. "Comparative Immunol. Microbiol. Infect. Dis. 2016, 48, 41–47, doi:10.1016/j.cimid.2016.07.006.
This review could easily double the information for those two sections, presumably this holds true for the other species in the review (chickens, cattle, horses etc). The authors must be more extensive in their literature search for this review to be acceptable for publication. Perhaps they should also mention non-oral cytokines for the purposes of completeness and whether these might be formulated for oral use, for example feline IFN omega.
Thank you for your suggestion, we deepen the literature search and we mentioned the non-oral cytokines (by adding a new section: 4. Regional vs. systemic effects of Type I IFNs, at line 122) and the possibility to use some of them orally (lines 365-372).
Minor Issues:
Line 50-53: this sentence is unclear and probably considered a "run on" sentence
Thank you for your suggestion, we re-phrase the sentence as follows: “Cytokines are also investigated in veterinary medicine at first for diagnostic purposes, and then also employed as a valid alternative to the large use of antibiotics in farm animals, in order to reduce resistance to antibiotics also used in human medicine [12]”.
Lines 199-200: Which IFN genes are stimulated? Be specific
The list of IFN stimulated genes (ISGs) that are modulated and presented in the reference 77 (Namangala et al. 2006) is very long. So we added in the text only the 5 most upregulated ones. Here the list.
IFN-α upregulated genes:
- Lysozyme C-2 precursor
- Chemokine orphan receptor 1
- Immediate early response 5
- Phagocytic glycoprotein 1 (CD44 antigen precursor)
- CDT1 protein
- Transcription factorlike protein MRGX
- DEAD-box protein 25
- Death effector domain-associated factor
- IFN-induced protein 44
- T cell surface glycoprotein CD1d precursor
- NOSTRIN protein
- Neutrophil elastase
- Histocompatibility 28
- CD42 effector protein 1
- Proteasome alpha 3 subunit
- Protein tyrosine kinase
- T-complex protein 1
- C-C chemokine receptor type 11
- Blood plasma glutamate carboxypeptidase precursor
- DQA protein precursor
- Heat-shock 70 kDa protein 12B
- Natural killer cell-specific antigen KLIP1
- MHC class II DM alpha-chain precursor
- T-lymphocyte maturation-associated protein
- Heat-shock 70-kilodalton protein 1°
- Leukocyte differentiation antigen CD84 isoform CD84a
- Programmed cell death protein 11
- Transferrin receptor 2
- Calpain small subunit 1
- Tyrosine-protein kinase 2 precursor
- Transcription factor (TFIIIC) alpha chain
- RNA helicase-DEAD box protein RH116
- B-cell receptor-associated protein 29
- Stress-activated protein kinase NJK3
- Complement component 2
- Cytochrome B-245 heavy chain
- Lymphocyte antigen 6 complex locus G6C protein precursor
- CD63 protein
- Platelet-activating factor acetylhydrolase precursor
- Thymus expressed novel gene-3 protein
- Heat-shock protein beta-3
- HLA-B-associated transcript 5
- Tumor necrosis factor receptor associated factor 6
- Apoptosis response zinc finger protein
- Brain stress early protein
- Oxygen-regulated protein precursor
- Tumor necrosis factor ligand superfamily member 5
- Secretory pathway component Sec 31B-1
IFN-α downregulated genes
- Tmem8 protein
- HMGCS1 protein
- Immunoglobulin VJC-region
- Uncharacterized hematopoietic stem/progenitor cell protein MDS029
- Cathepsin W precursor
- Transportin
- GW112 protein
- Superoxide dismutase copper chaperone
- Cross-immune reaction antigen PCIA1
- Conserved oligomeric Golgi complex component 1
- Adipose tissue-specific protein adipo Q
- BOVCZSD
- LP2254
- BM-011 protein
- Beta-adaptin 1
- T-cell receptor CD3 epsilon chain
- GrpE protein homolog 1
- STE20-like kinase
The word "showed" is used repeatedly, better choices would be "reported, observed, demonstrated"
Thank you for your suggestion, we replaced “showed” with other words all over the manuscript.